# Synthesis, Optical and Electrical Characterization of Amino-alcohol Based Sol-gel Hybrid Materials

**DOI:** 10.3390/polym12112671

**Published:** 2020-11-12

**Authors:** Bárbara R. Gomes, Rita B. Figueira, Susana P. G. Costa, M. Manuela M. Raposo, Carlos J. R. Silva

**Affiliations:** Centro de Química, Universidade do Minho, Campus de Gualtar, 4710-057 Braga, Portugal; barbara.sgomes11@gmail.com (B.R.G.); spc@quimica.uminho.pt (S.P.G.C.); mfox@quimica.uminho.pt (M.M.M.R.)

**Keywords:** hybrids, sol-gel, materials, amino-alcohol

## Abstract

This manuscript describes the synthesis and characterization of five new organic–inorganic hybrid (OIH) sol-gel materials that were obtained from a functionalized siloxane 3-glycidoxypropyltrimethoxysilane (GPTMS) by the reaction with the new Jeffamine^®^, namely three different diamines, i.e., EDR-148, RFD-270, and THF-170, a secondary diamine, i.e., SD-2001, and a triamine, i.e., T-403. The OIH sol-gel materials were characterized by UV-visible absorption spectrophotometry, steady-state photoluminescence spectroscopy, and electrochemical impedance spectroscopy. The reported OIH sol-gel materials showed that, with the exception of the samples prepared with Jeffamine^®^ SD-2001, the transmittance values ranged between 61% and 79%. Regarding the capacitance data, the values reported changed between 0.008 and 0.013 nF cm^−2^. Due to their optical and electrical properties these new OIH materials show promising properties for applications as support films in an optical sensor area such as fiber sensor devices. Studies to assess the chemical stability of the OIH materials in contact with cement pastes after 7, 14, and 28 days were also performed. The samples prepared with THF–170 and GPTMS, when compared to the samples prepared with RFD-270 and T-403, exhibited improved behavior in the cement paste (alkaline environment), showing promising properties for application as support film in optical fiber sensors in the civil engineering field.

## 1. Introduction

In the last few decades, the design and development of organic–inorganic hybrid (OIH) sol-gel materials for a wide range of applications has achieved a high scientific proficiency level. Several OIH materials have been reported for applications such as coatings for corrosion mitigation, smart windows, photochromic and electrochromic materials, sensors, optical filters, and absorbers, among others. The sol-gel method is recognized as a green, low-cost, and versatile route that allows organic–inorganic hybrid (OIH) materials to be obtained in a simple way by the reaction between polyetheramines and organoalkoxysilanes. The most remarkable properties of such OIH materials include chemical stability, suitable dielectric properties, selective ion binding, and ion conducting ability, depending on their organic and inorganic components. Such materials have also been reported as interesting luminescent sources [1,2,3,4,5] and hosts of luminescent species [6,7,8,9,10,11,12,13]. For instance, Severo Rodembusch et al. reported for the first time the synthesis of fluorescent hybrid aerogels in the blue–green–yellow region. The dyes were obtained by the reaction of amino benzazole derivatives with 3-(triethoxysilyl)propyl isocyanate. As precursor a pre-polymerized tetraethoxysilane was used [8]. In 2015, Brito et al. reported the entrapment of blue–green luminescent C-dots in transparent silica. The silica was obtained by pyrolysis of the methyl groups present in the nanometric silica grains [12].

Other materials such as siloxane-polyether OIH materials, also known as di- or tri-ureasil, have also been widely reported [11,14,15,16,17,18,19,20,21,22,23,24,25,26] for an extensive range of applications. The most common strategy to prepare such materials with the sol-gel method involves several hydrolysis and polycondensation reactions allowing their morphology, structure, and chemical composition to be tuned. For instance, Boev et al. [15] reported a simple procedure for the synthesis of ureasilicate materials. The authors studied the influence of the molar ratio between 3-isocyanate propyltriethoxysilane (ICPTES) and Jeffamine^®^ ED-600. A homogeneous and flexible material was attained and it was concluded that by changing the catalyst nature and the molar ratio of ICPTES/Jeffamine^®^ highly transparent samples within the visible range with dissimilar elastic properties were obtained. Later, the studied materials were doped with different nanoparticles, including CdS [16,26,27,28], CdSe [20,29], Zn_x_Cd_1−x_S [17], and PbS [30]. Ureasilicate materials have also been reported as coatings for corrosion mitigation in alkaline environments [19,22,31] and in the presence of chloride ions [32,33], showing good performance for the desired purpose. These materials were also reported as films for the development of a relative humidity (RH) optical fiber sensor [23]. The ureasilicate material was deposited by dip coating the fiber and studied in the range of 5% to 95% of RH. It was also concluded that when compared to other polymer-based solutions, the proposed material showed enhanced durability and sensitivity (22.2 pm/% RH) in monitoring the RH of two concrete blocks for one year [23]. Willis-Fox et al. [34] reported a strategy for the preparation of a conjugated polymer-di-ureasil composite material showing a tunable emission color from blue to yellow using a simple sol-gel method. Moreira et al. [18] reported the synthesis of ureasilicates and amino-alcohol silicate OIH sol-gel materials. The amino-alcohol silicates were obtained by reacting Jeffamine^®^ ED-600 or Jeffamine^®^ ED-900 with 3-glycidoxypropyltrimethoxysilane (GPTMS) using a molar ratio of GPTMS:Jeffamine^®^ = 2:1. The GPTMS precursor is a suitable molecule establishing amino-alcohol bonds between the inorganic (silicate) and organic (Jeffamine) components (i.e., a polyether chain) similar to that reported for the ureasilicate matrix [18]. The authors obtained flexible xerogels with high transparency within the visible range and showed that the amino-alcohol silicate samples exhibited suitable optical and mechanical properties that can lead to the production of low-cost and variable shape diffraction lenses in a wide range of substrates for optical applications [18]. Figueira et al. also reported the use of amino-alcohol silicate matrices as coating materials for corrosion mitigation of galvanized steel in an alkaline environment [22,35].

In the last few decades the use of optical fiber sensors (OFS) to monitor physical parameters of structures in the civil engineering field has been widely employed. For instance, the application of OFS to monitor properties such as the curvature and deflection of bridges [36,37,38,39], the onset of cracking [40,41,42], and temperature [43,44] has been consistently reported. Nevertheless, besides physical properties, other parameters such as pH level [45,46], relative humidity [47,48,49], chloride content [50,51,52], and alkali-silica reactions (ASR) [53] are also responsible for earlier failure of civil engineering structures. The development of robust and accurate monitoring sensing systems is of extreme importance as they will allow the service life of concrete structures to increase and costs saved at the same time. The monitoring of concrete structures using functionalized optical fiber sensors (OFS) with sol-gel films revealed to be an interesting approach to increase its service life [54]. 

Considering the steadily increasing attention being paid to these OIH films together with their promising properties, further research is needed. Therefore, this work reports the synthesis and optical and electrochemical characterization of new OIH amino-alcohol silicate-based materials for potential application on an OFS for health monitoring of concrete structures. In this manuscript, the new Jeffamines^®^ will be considered. To the authors’ knowledge no studies using the Jeffamine^®^ EDR-148, Jeffamine^®^ RFD-270, Jeffamine^®^ THF-170, Jeffamine^®^ SD-2001, and Jeffamine^®^ T-403 have been reported. Therefore, three different diamines (Jeffamine^®^ EDR-148, Jeffamine^®^ RFD-270, and Jeffamine^®^ THF-170), a secondary diamine (Jeffamine^®^ SD-2001), and a triamine (Jeffamine^®^ T-403) were made to react with GPTMS-producing amino-alcohol silicate matrices. More specifically, five new different OIH sol-gel materials were synthesized. 

To the authors’ knowledge no studies have been reported on the synthesis of OIH xerogel films using GPTMS and Jeffamine^®^ EDR-148, Jeffamine^®^ RFD-270, or Jeffamine^®^ THF-170 as a siloxane precursor. Concerning Jeffamine^®^ T-403, this precursor has been reported by different authors for a wide range of applications [55,56,57,58,59]. For instance, Spírkpvá et al. in 2003 reported the synthesis and structural and mechanical properties of the OIHs obtained by the reaction between GPTMS and three different Jeffamines^®^: T-403, D-230, and D-400 [55]. Nevertheless, the authors used different ratios and synthesis parameters [55].

The OIH sol-gel materials synthesized were characterized by UV–visible spectrophotometry, steady-state photoluminescence spectroscopy, and electrochemical impedance spectroscopy. The reported OIH sol-gel materials show interesting optical properties for application as a support matrix in OFS and the capacitance values changed between 0.008 and 0.013 nF cm^−2^. Preliminary studies were also conducted to assess the chemical stability of the OIH materials in contact with cement pastes for 7, 14, and 28 days. The samples prepared with Jeffamine^®^ THF-170 and GPTMS, when compared to the others, exhibited improved and promising properties in a cement paste environment.

## 2. Experimental

### 2.1. Reagents

Figure 1 shows the structure and acronym (in bold) of the precursors used in the synthesis of the OIH sol-gel materials. The structure of Jeffamine^®^ THF-170 (*Poly(oxy-1,4-butanediyl),α-hydro-Ω-hydroxy-,polymer with ammonia*) (Huntsman Corporation, Pamplona, Spain), hereafter referred to as THF-170, was not provided by the supplier. The structure which is indicated in Figure 1 is based on the information reported by Létoffé et al. for a similar Jeffamine^®^ (Jeffamine^®^ THF-100) [60]. Five different Jeffamines^®^ (structurally identified as polyetheramines) were kindly supplied by Huntsman, with different molecular weights and reactivity. Jeffamine^®^ EDR-148 is the most reactive Jeffamine^®^ when compared to diamines and triamines due to the unhindered nature of the amine groups (vide Figure 1).

Jeffamine^®^ RFD-270 is an amine containing both rigid (cycloaliphatic) and flexible (etheramine) moieties in the same molecule and the acronym RFD stands for “rigiflex diamine” (vide Figure 1). Jeffamine^®^ T-403 is a triamine prepared by the reaction of propylene oxide with a triol initiator, followed by the amination of the terminal hydroxyl groups (vide Figure 1). The Jeffamine^®^ THF-170 is based on a poly(tetramethylene ether glycol)]/(poly(propylene glycol) copolymer. Jeffamine^®^ SD-2001 is a polyetheramine, where SD stands for “secondary diamine.” The secondary amine groups provide a much slower reaction compared to primary amine groups. 

All the precursors used, namely the Jeffamines^®^ and the GPTMS (97%, Sigma-Aldrich, St. Louis, MO, USA), were used as supplied. As solvents, tetrahydrofuran (99.5% stabilized with ~ 300 ppm of BHT, Panreac, Darmstadt, Germany) and ultra-pure water with high resistivity (higher than 18 MΩ cm) obtained from a Millipore water purification system (Milli-Q®, Merck KGaA, Darmstadt, Germany) were employed. 

### 2.2. Synthesis Procedure of Xerogel Films 

The obtained xerogel films were synthesized in sequential steps, as schematized in Scheme 1.

Before the 3-glycidoxypropyltrimethoxysilane (GPTMS) addition, the Jeffamines^®^ were solubilized using tetrahydrofuran. After 10 min, the GPTMS was added to the Jeffamine^®^ (EDR-148, RFD-270, T-403, THF-170, or SD-2001) using a molar ratio of 2 GPTMS:1 Jeffamine^®^ in a glass vessel under stirring at 700 rpm for 20 min, with the exception of Jeffamine^®^ RFD-270, which was prepared with a molar ratio of 1 GPTMS:1 Jeffamine^®^. 

The reaction between the amine end group (–NH_2_) of the Jeffamine^®^ with different molecular weight and structure (Jeffamine^®^ EDR-148, Jeffamine^®^ RFD-270, Jeffamine^®^ T-403, Jeffamine^®^ THF-170, and Jeffamine^®^ SD-2001) and the epoxy group of GPTMS led to the formation of the OIH precursors of the future gel matrices that are named as conventional amino-alcohol silicates and referred to as A(148), A(270), A(403), A(170), and A(2001). The numbers in parentheses stand for the Jeffamine^®^ used, with the reference names indicated in Figure 1. The second step included the addition of H_2_O to obtain an H_2_O:Jeffamine^®^ molar ratio equal to 5.94. The mixture was stirred for another 10 min and placed into a Teflon^®^ mold (DuPont, Wilmington, DE, USA) and sealed with Parafilm^®^ that was pin-holed and placed in an oven (UNB 200, Memmert, Buechenbach, Germany) and kept at 40 °C for 15 days. This procedure ensures precise control and reproducibility of the conditions of hydrolysis/condensation reactions as well as the evaporation of the residual solvents. Scheme 1 shows that the implemented preparation conditions applied allowed homogeneous and transparent film samples free of cracks to be obtained.

### 2.3. OIH Sol-Gel Film Characterization

The OIH materials synthesized were characterized optically and electrochemically. All the measurements were conducted at room temperature. The UV–visible transmission and absorption spectra of the OIH film samples were obtained using a spectrophotometer (UV-2501 PC, Shimadzu, Duisburg, Germany) in the range of 250–700 nm. The fluorescence spectra were obtained using a spectrofluorometer (Fluoromax–4, Horiba Jovin Yvon, Madrid, Spain). The emission spectra were recorded in the wavelength range of 300–700 nm using different excitation wavelengths and acquired at front-face geometry at room temperature. 

Electrochemical impedance spectroscopy (EIS) measurements were carried out to characterize resistance, electrical conductivity, and the electric permittivity of the prepared OIH films. Measurements were performed using a two-electrode system in which the film was placed between two parallel Au disc electrodes (10 mm diameter and 250 μm thickness) and a support cell [33]. Measurements were performed in a Faraday cage using a potentiostat/galvanostat/ZRA (Reference 600+, Gamry Instruments, Warminster, PA, USA) by applying a 10 mV (peak-to-peak, sinusoidal) electrical potential within a frequency range from 1 × 10^5^ Hz to 0.01 Hz (10 points per decade) at open circuit potential. The frequency response data were displayed in a Nyquist plot. The Gamry ESA410 Data Acquisition software was used for data fitting purposes.

### 2.4. Preparation of Cement Paste

The chemical stability of the different OIH films was studied in a cement paste. The cement paste was prepared using cement type I 42,5R and distilled water. The ratio of water to cement (w/c) used was equal to 0.5. The OIH films used were 10 mm in diameter and the thickness ranged between 1.4 and 1.8 mm.

## 3. Results and Discussion

### 3.1. UV-Visible Spectrophotometry Analysis

Figure 2 shows the optical transmittance spectra as a function of wavelength obtained for the synthesized xerogel films A(148), A(270), A(403), A(170), and A(2001). The A(2001) samples show the lowest transmittance, followed by A(148), A(170), A(270), and A(403), with A(403) showing the highest transmittance values of between 400 and 700 nm. 

In the UV region between 250 and 300 nm, all the samples showed low transmittance, which is in accordance with the literature [18]. The transmittance data shows that at 400 nm wavelength, the xerogel that provided the highest value was A(403) with around 79%. At the same wavelength, A(270) showed a transmittance of around 74% and the lowest transmittance was given by A(2001) with a value of around 26%. At 400 nm, A(148) and A(170) showed a transmittance of 61% and 63%, respectively. As the wavelength increases the transmittance values increase, with A(403) reaching the highest value, i.e., 89% at 600 nm.

These results are in agreement with the results reported by Moreira et al. [18] and by Erdem et al. [61]. Both authors reported that the OIHs obtained with lower molecular weights show higher transmittance than films prepared with higher molecular weights. Moreira et al. [18] described the reaction between Jeffamine^®^ ED-600 and GPTMS as showing higher transmittance than films prepared by the reaction between Jeffamine^®^ ED-900 and GPTMS. This behavior suggests that the OIH transmittance is related to the molecular weight of the Jeffamine^®^. Another explanation may be related to the structure involved together with the interactions established between the Jeffamine^®^ and the GPTMS. Zea Bermudez et al. reported a study focused on the synthesis and FTIR characterization of OIHs using as precursors Jeffamines^®^ ED-600, ED-900, and ED-2000 with 3-isocyanatepropyltriethoxysilane. It was reported that the FTIR spectrum of U(2000) showed that the polyether chains of the parent diamine become less ordered upon incorporation into the inorganic backbone. The number of oxyethylene units present affected the amide I and amide II bands. This indicated that the N−H groups of the urea linkage were involved in the hydrogen bonds of different strengths. Moreover, the authors proposed the existence of non-hydrogen-bonded urea groups and hydrogen-bonded urea−urea and urea−polyether associations. On one hand the formation of urea−urea structures was favored in the U(600), while on the other hand the number of free carbonyl groups was higher in U(2000) [14]. Similar behavior for the A(2001) may be expected, which may justify the huge difference of transmittance between A(148), A(170), A(270), and A(403) since Jeffamine^®^ SD-2001 is a difunctional secondary amine derived from Jeffamine^®^ ED-2000. Nevertheless, since in this case GPTMS was used instead of 3-isocyanatepropyltriethoxysilane no further conclusions can be drawn. Therefore, further studies, namely FTIR analysis, should be conducted in order to clarify the interactions between the precursors. 

The thickness of the films ranged between 0.913 mm and 1.434 mm. The thickness obtained for the OIHs for A(148), A(270), A(403), A(170), and A(2001) was 1.301 mm, 0.913 mm, 1.051 mm, 1.434 mm, and 1.024 mm, respectively. Therefore, the absorbance measurements were divided by the thickness obtained for each film. Figure 3 shows the UV-vis absorption spectra normalized to the film thickness of the four synthesized xerogel films (A(148), A(270), A(403), A(170), and A(2001)).

The highest absorbance peaks were found in the UV region between 250 and 300 nm (Figure 3). The maximum absorption wavelength obtained for each xerogel film for A(148), A(270), A(403), A(170), and A(2001) was 294 nm, 290 nm, 281 nm, 276 nm, and 310 nm, respectively. Regardless of the structure of the Jeffamine^®^ used, as the molecular weight of Jeffamine^®^ increases, the maximum wavelength in the UV region (between 250–300 nm) decreases. An exception was found for A(2001), which showed the highest molecular weight and a maximum at a wavelength of around 310 nm. The highest and the lowest absorbance peaks were obtained for A(2001) and A(403) xerogel films. Figure 3 also shows that, with the exception of A(2001), all the other OIH matrices are transparent from 300 nm forward. Therefore, these materials are suitable for probe immobilization in the range between 300 and 700 nm. Considering the low transmittance obtained for A(2001) and its translucency, no further studies were conducted in characterizing A(2001). Moreover, previous studies showed that as the molecular weight of a Jeffamine^®^ increases, the resistance of the OIHs to a highly alkaline environment decreases [22,32].

### 3.2. Photoluminescence Spectrophotometry Analysis

Figure 4 shows the emission fluorescence spectra obtained for the synthesized xerogel films A(148), A(270), A(403), and A(170) at different excitation wavelengths. The limits of excitation wavelengths were defined according to the maximum wavelength recorded for each OIH film in UV-vis analysis, always starting at 250 nm. 

Figure 4 shows that the OIH materials show an intrinsic emission. In the case of the A(270) and A(170) OIH matrices, it can be observed that the wavelength of the emission peak displaces to higher wavelengths when the excitation wavelength increases. This intrinsic emission linked to these OIH matrices is due to the photoinduced proton transfer between defects NH_3_^+^/NH^−^ and due to the electron−hole recombination occurring in the siloxane nanoclusters [62]. The emission wavelength dependency with the excitation energy is connected to unorganized processes that are generally linked to transitions that occur between localized states in non-crystalline structures [62]. According to Carlos et al. [2], in OIHs similar to the ones reported here the hierarchy in the silica backbone dimension defines the emission wavelength. This is in agreement with the findings reported here. This energy dependence of the emission wavelength with the excitation energy is related to the size of the silica clusters [63]. The same authors reported that larger clusters emit at longer wavelengths than smaller clusters. Generally, shorter polyether chains (A(148) and A(403)) provide samples with higher photoluminescence while larger chains may induce a dilution effect that may reduce the luminescence efficiency [63], which is in agreement with the data reported here. In the case of A(148) and A(403) it can be observed that the wavelength of the emission peak displaces to lower wavelengths when the excitation wavelength increases. Photoluminescence spectra of the samples with shorter polymer chains such as the ones used in A(148) and A(403) suggest that the electron−hole recombination occurring in the siloxane nanoclusters is large enough to allow efficient energy-transfer mechanisms. 

The full width at half maximum did not decrease with decreasing excitation energy. Considering that the OIH materials are a biphasic system (i.e., organic and inorganic components are mixed at nanometric scale), this may contribute to a higher light scattering that may lead to changes within the sample and to a general broadening of the peaks. Generally, Figure 4 shows that the full width at half maximum did not decrease with decreasing excitation energy.

Figure 5 shows the photographs of the xerogel samples synthesized and their photoluminescence response when excited with UV light, i.e., an excitation wavelength of 365 nm. 

The blue color of the A(148), A(270), A(403), and A(170) xerogels after excitation with UV radiation arises from the photoluminescence emission in the 450–470 nm region [29]. It is generally accepted that silicon-based materials can emit light within a wide energy region that ranges from ultraviolet to infrared [2,64,65]. The backbone silicon-based structures are responsible for the emission energy, i.e., an increase in the siliceous network may result in a decrease in the corresponding energy gap [2]. The emission energy of silicon-based materials depends on the hierarchy of their backbone dimensions. Since the network dimension changes, the band gap energies change accordingly. This dependence of the energy gap on the backbone dimensions is related to the extension of the silicon σ-conjugations through the OIH network. These induce the delocalization of the electrons, leading to the formation of electronic band structures. Increasing their extension along the silicon backbone induces a decrease in the energy gap values, with a corresponding increase in the skeleton dimensions. Moreover, the Si–O–Si network is considered responsible for the blue emission of oxidized porous silicon [2].

### 3.3. EIS Measurements

Electrochemical impedance spectroscopy (EIS) is a very interesting technique widely used for the characterization of OIH sol-gel materials [32,33,35,66,67]. Figure 6 shows the Nyquist plots of the pure OIH films based on A(270) (Figure 6a), A(403) (Figure 6b), A(2001) (Figure 6c), and A(170) (Figure 6d) matrices, correspondingly. The experimental and the fitting results are shown in the Nyquist plots (Figure 6) and are schematized by squares and a continuous line, respectively. The equivalent electrical circuits (EEC) used for each sample were introduced as an inset in each Nyquist plot.

The Nyquist plots illustrated in Figure 6 show that generally in the region of high frequencies a semicircle intersects the x-axis. It can also be observed that the diameter of the semicircle changes with the Jeffamine^®^ used. This indicates that the dielectric properties of the OIH sol-gel materials (e.g., conductivity, capacitance, etc.) change with the different Jeffamine^®^ used in the OIH synthesis. The data obtained at lower frequencies (Figure 6d) describes a line suggesting another electrochemical process that can be assigned to the interfacial phenomena between the gold electrodes and the OIH disc, which is in accordance with the literature [35]. However, this particular phenomenon will not be considered in the EIS analysis since it is not necessary to characterize the dielectric properties of the materials that are being studied. The EIS results regarding OIH A(148) films are not reported since it was not possible to perform its measurements. This may be due to the high rigidity and the difficulty in establishing the contact between the gold electrodes and the A(148) OIH discs. The high rigidity contributes to a poor electric contact between the OIH film and the gold electrode discs, disabling the electrical response. 

The Nyquist plots (Figure 6) show a depressed form, therefore the analysis of all the impedance responses was based on an EEC where constant phase elements (CPE) were used in place of pure capacitance. 

As reported, the impedance of a CPE can be defined as [68]:(1)$$Z_{\mathrm{CPE}}=\frac{1}{\left[Q{\left(j\omega \right)}^{\alpha }\right]}$$

In Equation (1), both Q and α parameters are independent of the frequency. When α = 1, Q stands for the capacity of the interface [67]. However, when 0 < α < 1 the system shows behavior that is generally linked to surface heterogeneity and the impedance for the EEC is given by Equation (2) [68]: (2)$$Z_{\mathrm{CPE}}=\frac{R_{\mathrm{sample}}}{\left[1+{\left(j\omega \right)}^{\alpha }{\mathrm{QR}}_{\mathrm{sample}}\right]}$$

The interfacial capacitance (C_eff_) is given by Equation (3) using the estimated Q value [68]:(3)$$C_{\mathrm{eff}}={\left[{\mathrm{QR}}_{\mathrm{sample}}{}^{\left(1-\alpha \right)}\right]}^{\frac{1}{\alpha }}$$

Table 1 shows the values of the proposed EEC elements obtained from the EIS data fitting for all the OIH materials synthesized. 

Table 1 shows that all the resistances of the new OIH film samples synthesized, namely A(170), A(270), A(403), and A(2001), are between 10^7^ and 10^10^ Ω. With the exception of the values obtained for A(400) [35] (vide Table 1), all the new synthesized OIH materials show higher values when compared to the ones already reported for A(900) and A(2000).

The values obtained for the elements of the EEC proposed and reported in Table 1, i.e., the values of resistance (R_Sample_), the constant phase element (CPE), and α, were used to obtain the C_eff_ using Equation (3). The resistance (R) and capacitance (C) values were normalized to cell geometry dimensions and calculated using Equations (4) and (5), respectively. The conductivity (σ) and relative permittivity (ε_r_) were also determined using Equations (6) and (7), respectively. The values were obtained using the equations below, where A_Au_ is the area of the gold electrodes, d_Sample_ is the thickness of the analyzed OIH film sample, and ε_0_ stands for the vacuum permittivity in nF cm^−1^:(4)$$R=R_{\mathrm{sample}}\times A_{\mathrm{Au}\;\mathrm{disc}}$$
(5)$$C=\frac{C_{\mathrm{eff}}}{A_{\mathrm{Au}\;\mathrm{disc}}}$$
(6)$$\sigma =\frac{d_{\mathrm{sample}}}{A_{\mathrm{Au}\;\mathrm{disc}}}/R_{\mathrm{sample}}$$
(7)$$\varepsilon_{r}=\frac{C_{\mathrm{eff}}\times d_{\mathrm{sample}}}{\varepsilon_{0}}\times A_{\mathrm{Au}\;\mathrm{disc}}$$

The information regarding the electrical properties of the OIH films produced, including normalized resistance (R), capacitance (C), conductivity (σ), and relative permittivity (ε_r_), is displayed in Table 2.

Table 2 shows that for the OIHs A(270), A(400), A(403), A(600), A(900), and A(2000), as the molecular weight of the Jeffamines^®^ used in the synthesis increases, the normalized resistance (R) decreases, which is in accordance with the literature [22,35]. The difference obtained for the resistance of A(270) compared to A(400) may be due to the fact that Jeffamine^®^ RFD-270 is an aliphatic amine containing both rigid (cycloaliphatic) and flexible (etheramine) segments in the same molecule (vide Figure 1). In the case of A(400), the Jeffamine^®^ used in the synthesis (Jeffamine^®^ D-400) is a difunctional primary amine. This polyetheramine is characterized by repeating oxypropylene units in the backbone. This difference between the backbones of the Jeffamines^®^ used (i.e., RFD-270 and D-400) may explain the two orders of magnitude difference found between the resistance of A(270) and A(400) and the capacitance that in case of A(400) is 4.375 times higher than A(270). Regarding the A(170) samples, a lower resistance was expected considering the molecular weight of the Jeffamine^®^ used (1700 g mol^−1^). Previous publications showed that as the molecular weight of the Jeffamine^®^ increases, the resistance of the OIHs decreases due to the organic component increase [22,35]. Jeffamine^®^ THF-170 is chemically based on a [poly(tetramethylene ether glycol)]/(poly(propylene glycol)) copolymer that contains a significant amount of secondary as well as primary amines. Therefore, this might be the main reason for the high resistance obtained. The main difference between the resistance obtained for A(2000) and A(2001) can be explained by the disparity between the Jeffamine^®^ used in each synthesis. A(2001) was synthesized using a difunctional secondary amine (Jeffamine^®^ SD-2001) while A(2000) was obtained using a polyether diamine based predominantly on a polyethylene glycol (PEG) backbone. Jeffamine^®^ SD-2001 is a difunctional secondary amine derived from the Jeffamine^®^ D-2000 amine, which are polyether diamines based on a poly(propylene glycol) (PPG) backbone. This may explain the main differences found between A(2000) and A(2001) regarding the electrical properties (Table 2) in which the resistance of A(2001) increases 2.04 times compared to the resistance obtained for A(2000).

The new OIH materials synthesized show a ε_r_ between 8 and 24, with A(170) and A(2001) showing the lowest and the highest values, respectively. The capacitance values obtained are between 0.008 and 0.013 nF cm^−2^, with A(170) and A(2001) showing the lowest and the highest, respectively. Moreover, the new amino-alcohol-based sol-gel materials show lower ε_r_ and capacitance values than the ones already reported.

### 3.4. Preliminary Assessment of the OIH in Contact with Cement Paste 

A preliminary assessment of the OIH materials in contact with a cement paste prepared with a w/c = 0.5 was conducted. The preliminary tests were only performed for the OIHs with the highest transmittance values, i.e., A(403), A(270), and A(170). The main objective of these studies was to assess if these new OIHs were stable and resistant when in contact with a high alkaline environment. A cement paste with a w/c = 0.5 was chosen to mimic the concrete alkalinity. The samples were immersed in the cement paste for 7, 14, and for 28 days. After 7, 14, and 28 days the cured paste samples were broken to check the physical aspect of the samples. Figure 7 shows the samples A(270), A(403), and A(170) after 7 days of being embedded in the mentioned cement paste.

Figure 8 shows the results obtained for the OIH samples after 28 days of being embedded in the cement paste. 

After 7 days of contact with the cement paste (Figure 7), except for OIH sample A(170), OIH A(270) and A(403) suffered severe degradation and reacted with the cement paste. Even though sample A(403) seems to be less resistant than A(270), due to the alkalinity of the cement pastes, it was not possible to extract OIH samples A(270) and A(403) from the cement paste after 7 days, as can be observed in Figure 7. Sample A(170) appears to be resistant to the high alkaline environment of the cement paste and did not experience visible degradation. It can be observed that after 28 days the same behavior was found. OIH sample A(170) showed improved stability when compared to the other samples. The promising preliminary results suggest that further tests such as EIS should be conducted. By performing EIS measurements, i.e., assessing resistance and capacitance of the A(170) samples before and after immersion in the cement pastes, it will be possible to quantify the degradation impact of the alkaline environment.

Considering the results obtained, the thickness of the OIH A(170) films was measured before and after 7, 14, and 28 days of being embedded in the cement paste. Table 3 shows the thicknesses obtained for each A(170) OIH film tested.

Table 3 shows that thickness decreases slightly over time for the A(170) samples. After 28 days a decrease of 0.22% of thickness was found. The highest decrease percentage was found for the OIH sample exposed for 7 days (around 3.17%), followed by the sample exposed for 14 days, which lost around 1.27% thickness. 

## 4. Conclusions

The synthesis of the five new OIH sol-gel materials, homogeneous and crack-free, based on amino-alcohol silicate matrices, namely A(148), A(270), A(403), A(170), and A(2001), has been reported. The results obtained allow for the conclusion that the different Jeffamines^®^ used influence the optical and the electrochemical properties. The OIH matrices with the lowest and highest transmittance, between 400 and 700 nm, were given by A(2001) and A(403), respectively. The photoluminescence spectra obtained for xerogel films A(148), A(270), A(403), and A(170) allow for the conclusion that these materials show an intrinsic emission and provide generally shorter polyether chains than samples with higher photoluminescence. The capacitance values obtained were lower than the ones already reported for similar materials and ranged between 0.008 and 0.013 nF cm^−2^. The A(170) samples showed good behavior in contact with cement paste, allowing for the conclusion that only A(170) samples are resistant to highly alkaline environments. Therefore, considering the optical and electrical properties and the chemical stability of A(170) samples in contact with cement paste, it can be concluded that these samples show promising properties as support films in optical sensor fields such as fiber sensor devices for application in the civil engineering field. Regarding the A(270) and A(403) samples, it can be concluded that, despite the interesting optical and electrical properties, these OIH samples are not suitable to be used in highly alkaline environments. Moreover, further studies should be conducted on the A(170) samples in order to quantify the degradation impact due to the alkaline environment.

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
