# Peer review of "Synthesis, Optical and Electrical Characterization of Amino-alcohol Based Sol-gel Hybrid Materials"

_polymers, 2020, doi:10.3390/polym12112671_

Round 1
Reviewer 1 Report
The manuscript “Synthesis, optical and electrical characterization of amino-alcohol based sol-gel hybrid materials” by Dr. Figueira et al. presents the synthesis and photophysical characterization of new materials from a functionalized siloxane 3-glycidoxypropyltrimethoxysilane (GPTMS) by the reaction with amino compounds. The manuscript is well-written, interesting, and fits with the journal scope. The spectroscopic characterization is well-done and sounds solid. I will indicate publication, but not in its present form. Before approving the publication I would like to address some questions/suggestions to the authors:
1) The abstract must be concise and highlight the main results. In this sense, I suggest removing the sentence "In the last few decades ... among others.", which in my opinion is more suitable in the introduction section. In addition, please present one sentence highlighting the interesting optical properties of these materials.
2) To broaden the background about hybrid materials as hosts of luminescent species (not only lanthanides) I suggest to see for example J. Mater. Chem., 2005,15, 1537-1541, Langmuir 2009, 25, 22, 13219–13223, Journal of Sol-Gel Science and Technology 52, 2009, 305–308, New J. Chem., 2012,36, 2506-2513 and Carbon 91, 2015, 234-240).
3) I strongly suggest to change table 1 into a Figure, since the last two columns are not necessary, only the chemical structures and chemical name/acronym
4) Figure 1 is not a figure but a scheme.
5) In Figure 2 please change "transmission" by "transmittance". It could be very interesting to present at least the images from film A (403) and A (2001) in this figure to better show the differences between them. The authors describe the values observed and related to previous results where the authors observed that "films obtained with lower molecular weight 183 (obtained by the reaction between Jeffamine® ED-600 and GPTMS) showed higher transmittance 184 than films prepared higher molecular weight polyether (obtained by the reaction between Jeffamine® 185 ED‐900 and GPTMS)". But why? What is going on at the molecular level to show such different behavior (for instance between A (403) and A (2001)). In addition, did the baseline from A (2001) correctly performed? Please see Figure 3, sample A (2001) where at 700 nm the baseline is already too high, which can be related to these observations from Figure 2.
6) Table 2 is not necessary. I suggest removing it. The same for Table 3 can be discussed in the text.
7) Figure 3 is not normalized (please correct the Y-axis) and also absorbance is dimensionless, so please remove "u.a."
8) Table 3 is also not necessary. Present these data in the text.
9) Based on the affirmation (Page 7, line 207) "Considering the low transmittance obtained for A(2001) no photoluminescence spectrophotometry analysis was performed", was based on the color of the material? Because low transmittance means higher absorbance, which could lead to fluorescence signal.
10) In Figure 7 I suggest increasing the letters inside the figure.
Reviewer 2 Report
This manuscript introduced about the This manuscript describes the synthesis and characterization of five new OIH sol-gel materials. Although the author has provided several experiment and explanations, there are some questions as follows.
- In Result 3.2, A170 and A270 explain why it is observed that as the wavelength increases, the wavelength of the emission peak shifts to a higher wavelength. However, there is no explanation for the displacement of 148 and 403 to lower wavelengths. If there are any theoretical and experimental results for this, it seems to have to be presented.
- In Fig. 5, color differences are not clearly distinguished in the presented picture. Can you suggest something that can distinguish this more clearly to support the manuscript?
- It seems to suggest the chemical stability of the substances presented in this study. Have you verified that these materials are environmentally stable? If it is possible to provide evidence for this, it would be better to mention it.
Reviewer 3 Report
The present work reports the preparation and properties of organic-inorganic hybrid sol-gel materials based on 3-glycidoxypropyltrimethoxysilane and a series of polymeric diamines.
The manuscript is clear, well-written and merits publication.
A few minor remarks:
i) Page 6. Two tables appear, named Table 3. In fact, the authors should try to merge somehow these tables into just one table.
ii) Page 9, lines 267-269: “The EIS…in the EIS cell.”. These phrases should be removed, since they are repeated just below.
iii) The authors should consider whether they should add the following references, for a more complete presentation of the work done in this area:
a) Science 1997, 276, 1826, b) Chemistry of Materials 1998, 10, 3777, c) Journal of Luminescence 2003, 101, 135.
Reviewer 4 Report
This manuscript reported an interesting work of Jeffamine and silicate hybrid materials. The experiments and data is comprehensive. After some revision, it is suggested to be accepted. The comments and suggestions are listed below.
- Line 33-36: not all the OIH materials have the suitable dielectrical properties, selective ion binding and ion conducting ability, depending on their organic and inorganic components.
- Line 38 and Line 42-43: why are siloxane-polyether OIH materials also known as di- or tri-ureasil? What does “urea” comes from? Based on Line 42-43, “urea” comes from the reaction between amino and ioscyanate groups? However, based on “Reagents” Section of this manuscript, no ICPTES was used.
- Line 54-56: Ref 27 has not the author of Willis-fox. Does “conjugated polymer” refer to the polymer with long pi-bond conjugated backbone? However, Ref 27 has nothing to do with such conjugated polymer.
- Line 61: what kind of chemical structure is “amino-alcohol bridge”?
- Line 75-77: this sentence is too long and complicate to follow. As well, “obtained using as siloxane precursor the GPTMS and as Jeffamines T-403 or D-230 or D-400” (Line 95 and 96) should be revised.
- In Table 1: “Polypropylene glycol” (as well in Line 336-337) should be “Poly(propylene glycol)”. As for Jeffamine®THF170, based on its name, its chemical structure can be given. As for Jeffamine® RFD-270, “poly” should be removed from “polyetheramine” (Line 115), since it just has one unit of ethylene glycol.
- Figure 1 and Line 131: “a stoichiometric ratio” should changed to “a molar ratio”. Jeffamine T403 has three primary amino groups and Jeffamine SD2001 has two secondary amino groups, being quite from other Jeffamines having two primary amino groups. Commonly, one primary amino group reacts two epoxy groups of GPTMS. Thus, it is so strange that “2:1 molar ratio” was used for all Jeffamines and “1 GPTMS : 1 Jeffamine” for Jeffamine® RFD-270.
- Line 165: clarify if the ratio is referred to weight ratio.
- Transmittance study and Figure 2: visible light is ranged in the wavelength of 390-780 nm. However, the highest wavelength reported in this manuscript is only 700 nm. The same issue is claimed to Figure 3 and absorption study.
- Line 189 and 190: the sentence should be revised.
- Figure 4: why were the starting and ending wavelengths different among four xerogel films?
- Line 225: as for this manuscript, what compound do “C=O groups” come from?
- Line 252: what is “s-conjugation”? Is it “sigma-conjugation”? What are “electronic band structures” (Line 253)?
- Title of Figure 5: “an excitation wavelength of 365 nm” may be “an excitation with the light with wavelength of 365 nm”. As well, the statement in the blanket of the title of Figure 6 should be revised.
- Table 4: the titles of the last two column were lost.
- Line 302: there maybe exists an type-error for “A(400)”. Something is lost after “the constant phase element (CPE) and” (Line 305).
- Line 328 and 329: This sentence is not narrated clearly. Why does the announced correlation between the resistance and Jeffamine MW exist?
- Figure 7: the samples for A(270) and A(403) are not clearly present.
- Line 384 and 385: since only A(170) “showed good behavior in contact with cement pastes”, “these samples” should be replaced with “only A(170)”.
